# Complete Pseudo-Anodontia in an Adult Woman with Pseudo-Hypoparathyroidism Type 1a: A New Additional Nonclassical Feature?

**DOI:** 10.3390/diagnostics12122997

**Published:** 2022-11-30

**Authors:** Salvatore Sciacchitano, Gian Paolo De Francesco, Maria Piane, Camilla Savio, Claudia De Vitis, Simona Petrucci, Valentina Salvati, Marina Goldoni, Marco Fabiani, Alvaro Mesoraca, Caterina Micolonghi, Barbara Torres, Annalisa Piccinetti, Roberto Pippi, Rita Mancini

**Affiliations:** 1Department of Clinical and Molecular Medicine, Sapienza University, Via di Grottarossa, 1035/1039, 00189 Rome, Italy; 2Department of Oncological Science, Breast Unit, St. Andrea University Hospital, Via di Grottarossa, 1035/39, 00189 Rome, Italy; 3UOD Medical Genetics and Advanced Cell Diagnostics, St. Andrea University Hospital, Via di Grottarossa, 1035/39, 00189 Rome, Italy; 4Scientific Direction, IRCCS Regina Elena National Cancer Institute, Via Elio Chianesi 53, 00144 Rome, Italy; 5Medical Genetics Division, Fondazione IRCCS Casa Sollievo della Sofferenza, 71013 San Giovanni Rotondo, Italy; 6ALTAMEDICA, Human Genetics, Viale Liegi 45, 00198 Rome, Italy; 7Department of Experimental Medicine, Faculty of Medicine and Dentistry, Policlinico Umberto I University Hospital, Sapienza University of Rome, 49971 Rome, Italy; 8Department of Internal Medicine, Belcolle Hospital, Str. Sammartinese, 01100 Viterbo, Italy; 9Department of Oral and Maxillofacial Sciences, Sapienza University, Via Caserta 6, 00161 Rome, Italy

**Keywords:** pseudo-anodontia, pseudo-hypoparathyroidism type 1a, PHPT-1a, *GNAS*

## Abstract

Pseudo-anodontia consists in the clinical, not radiographic, absence of teeth, due to failure in their eruption. It has been reported as part of an extremely rare syndrome, named GAPO syndrome. Pseudo-hypoparathyroidism type 1a (PHPT-1a) is a rare condition, characterized by resistance to the parathyroid hormone (PTH), as well as to many other hormones, and resulting in hypocalcemia, hyperphosphatemia, and elevated PTH. We report here the case of a 32-year-old woman with a long-standing history of non-treated hypocalcemia, in the context of an undiagnosed PHPT-1a. She had an intellectual disability, showed clinical features of the Albright hereditary osteodystrophy (AHO) and presented signs of multiple hormone resistances. She received treatment for seizures since the age of six. Examination of her mouth revealed a complete absence of teeth. Treatment of hypocalcemia and hormone deficiencies were started only at 29 years of age. Genetic testing demonstrated the presence of a frameshift variant in the *GNAS* gene in the proband as well as in her mother. A Single Nucleotide Polymorphism (SNP) array analysis failed to demonstrate pathogenic copy number variants (CNVs) but showed several regions with loss of heterozygosity (LOHs) for a final percentage of 1.75%, compatible with a fifth degree of relationship. Clinical exome sequencing (CES) ruled out any damaging variants in all the teeth agenesis-related genes. In conclusion, although we performed an extensive genetic analysis in search of possible additional gene alterations that could explain the presence of the peculiar phenotypic characteristics observed in our patient, we could not find any additional genetic defects. Our results suggest that the association of genetically confirmed PHPT-1a and complete pseudo-anodontia associated with persistent patchy alopecia areata is a new additional nonclassical feature related to the *GNAS* pathogenic variant.

## 1. Introduction

Anodontia (OMIM 206780) is a rare disorder characterized by the failure to develop all primary teeth by the age of 12 to 13 months or permanent teeth by the age of 10 years. Pseudo-anodontia is defined as the clinical, but not radiographic, absence of teeth due to a failure in their eruption. In these cases, radiographic examination allows the disclosure of the retained teeth in the jaws. The exact incidence of this condition is not known and it can be due to many different possible causes [1]. It has been reported as part of a rare syndrome, with only 60 cases reported so far, named GAPO syndrome (OMIM 230740) [2]. In addition to pseudo-anodontia, this syndrome is also characterized by growth retardation, alopecia and ocular manifestations. Primary failure of tooth eruption (PFE) of permanent teeth (OMIM 125350) is a rare disorder, associated with some syndromes involving skeletal development, but is also known as a non-syndromic autosomal dominant condition [3]. Pseudo-hypoparathyroidism type 1a (PHPT-1a) (OMIM: 103580) is a rare condition, too. Its exact prevalence is not known and it has been estimated at 1/295,000 in Japan and at 1/150,000 in Italy. It is characterized by a resistance to the parathyroid hormone (PTH), resulting in hypocalcemia, hyperphosphatemia, and elevated PTH. In addition, patients with this condition often develop resistance to other hormones that act through the common alpha subunit of the stimulatory G protein (Gsα) signaling pathway, including TSH, gonadotropins, growth-hormone-releasing hormone (GHRH) and α-melanocyte-stimulating hormone. Clinical features consist in a variety of manifestations, generally known as Albright hereditary osteodystrophy (AHO). Symptoms related to hypocalcemia include numbness, seizures, tetany, cataract or dental problems. This condition is part of the highly heterogeneous group of diseases caused by impairments in the parathyroid hormone (PTH) signaling pathway. They have been classified by the EuroPHP network under the common term ‘inactivating PTH/PTHrP signalling disorder’ (iPPSD) [4]. The common feature is, in fact, represented by an impairment in PTH and/or the PTHrP cAMP-mediated pathway. According to this new nomenclature, there are some criteria that are considered major in making this diagnosis. They include the resistance of the renal proximal tubule to the action of PTH, the presence of ectopic calcifications and the brachydactyly type E. Other minor criteria consist in TSH resistance, other hormonal resistances, motor and cognitive retardation or impairment, intrauterine and postnatal growth retardation, obesity/overweight, and flat nasal bridge and/or maxillar hypoplasia and/or round face. In approximately 70–80% of PHPT-1a, haploinsufficiency of *GNAS*, due to heterozygous-inactivating mutations in the maternally inherited allele (locus 20q13), can be detected [5,6]. Such mutations are scattered along the entire *GNAS* gene [7] and lead to a diminished stimulatory Gsα expression and/or function, resulting in AHO with multiple hormone resistances [8]. We report here a rare case of complete pseudo-anodontia which occurred in a female patient affected by PHPT-1a, showing some peculiar phenotypic characteristics. We performed an extensive genetic analysis in search of possible additional gene alterations that could explain the presence of the peculiar phenotypic characteristics observed in our patient.

## 2. Case Presentation

A 32-year-old woman, whose diagnosis of iPPSD was made at 29 years of age, came to our attention, accompanied by her mother. The hypocalcemia was not recognized nor treated until that age; therefore, she suffered from many complications related to both hypocalcemia and resistance to various hormones. The patient was born at 41 weeks of gestation by cesarean section. She had a normal weight at birth but quickly began to gain weight excessively, with a consequent early onset of obesity. In addition to obesity, the mother referred a history of motor and mental retardation and epilepsy in infancy, treated with carbamazepine and phenobarbital. Both parents (Dominicans in origin) were healthy. The patient was not treated with calcium until the age of 29 when hypocalcemia was detected and oral calcium supplementation was started. The patient experienced menarche at 11 years of age, but gonadotropin resistance resulted in the incomplete development of secondary sexual characteristics. At 15 years of age, she developed bilateral cataracts, and underwent cataract surgery. At 30 years of age, she underwent surgery for the removal of an ovarian serous cystadenoma, measuring 15 cm in diameter. At 29 years of age, the lab tests were remarkable for calcium of 6.7 mg/dL (8.4–10.2), phosphorus of 4.2 mg/dL (2.5–4.5) and magnesium of 1.5 mg/dL (1.5–2.6). The PTH was markedly elevated at 453 pg/mL (15–68), while the 25-hydroxy vitamin D was insufficient at 28 ng/mL (30–100), and the diagnosis of pseudohypoparathyroidism (PHP) was made. Three years later, she came to our attention and she was receiving supplementation with calcium 1 gr QD and with calcifediol 50 µg/dose at 1-week interval, and with levothyroxine at the dose of 50 µg QD, because of hypothyroidism due to TSH resistance. Treatment normalized the calcium level at 8.4 mg/dL (8.4–10.2), and the PTH was still high but reduced to 329 pg/mL (15–68), while the 25-hydroxy vitamin D was sufficient at 43 ng/mL (30–100). At the physical examination, the patient was conscious, cooperative and alert. The blood pressure was 103/71 mmHg and the pulse rate 81/min. Even if the serum calcium level was normal, the muscle showed generalized hypertonicity and stiffness, and the arms were hard to maneuver. She had a singular phenotype, showing the typical features of the AHO (Figure 1).

She was short-statured (height 143 cm) and overweight (weight 59.0 kg, BMI = 28.9 kg/m^2^), with a round face, a short neck, a broad chest and obese abdomen (Figure 1, panels a, b). Physical examination of the hands indicated a type E brachydactyly, that affected the 4th and 5th fingers more, and a type D brachydactyly (Figure 1, panel d). She had a positive knuckle sign (Archibald’s sign), consisting in the appearance of a dimple at the position of the 4th and 5th knuckles on clenching of the fists. Physical examination of the feet indicated brachydactyly, with shortening of the first toe in both feet as well as of the 4th metatarsal bone (Figure 1, panel e). Examination of the mouth showed that the upper lip was larger than normal and the lower lip was prominent (Figure 1, panel c). Intraoral examination revealed complete anodontia, broad, flat alveolar ridges, and shallow vestibules. The mucosa over the ridges was pale and atrophic with smooth, globular elevations along the ridges, probably implying that the teeth were present within the bone. The hair showed a persistent patchy alopecia areata that the patient was masquerading with hair extensions (Figure 1, panel f). An ultrasonographic examination of the thyroid gland and of the parathyroid did not show any alteration, and both sonography and Rx mammography of the breast revealed the presence of bilateral subcutaneous calcifications (data not shown). No optic atrophy was detected at fundoscopic examination. Moreover, no other ocular manifestations, typical of the GAPO syndrome, were detected, including glaucoma, strabismus, photophobia, megalocornea, myelinated retinal nerve fiber layer, keratoconus, nystagmus or ptosis. An X-ray of her hands and feet revealed significant shortening of the 4th and 5th metacarpals (Figure 2, panels c, d) and of the 4th metatarsal (Figure 2, panel e, f). An X-ray of the skull showed a copper beaten appearance and hyperostosis of the calvaria (Figure 2, panel a, b). Abnormal calcifications were also visible (Figure 2, panel b).

Dental panoramic radiography revealed peculiar alterations. In particular, bone anomalies were visible, with reduced height of the mandibular body, especially in molar regions, and reduced height of maxillary bones, especially in the anterior area, below the nasal cavities. Radiopaque areas appear to involve some teeth (specifically the upper and lower left first molars). Dental anomalies consisted in 16 abnormal un-erupted teeth in the maxilla and 12 in the mandible. It was not possible to define if they are deciduous or permanent. Many of these teeth had very short roots and many had no roots at all. Many of them were mal-positioned, with various degrees of inclination. No teeth were visible in the incisive area of the mandible. Almost all teeth appeared small and morphologically not-well-defined. Only a few of them showed a different degree of radiopacity between the superficial crown layer and the remaining part of the tooth. This was possibly due to no difference in the degree of mineralization between dentine/cement and enamel, or due to a lack of enamel (Figure 3).

### Treatment Outcome and Follow-Up

The patient is now receiving supplementation with calcium 1 gr BID and with calcifediol 50 µg/dose at 1-week interval. She is also receiving levothyroxine supplementation at the dosage of 50 µg QD. Treatment of epilepsy consists in administration of carbamazepine and phenobarbital. After one year of observation, before she returned to her country of origin, there was no change in the clinical condition of the patient.

## 3. Aim of the Study

Since the typical clinical presentation of PHPT-1a does not include complete pseudo-anodontia and alopecia, we decided to perform an extensive molecular genetic analysis in search of the molecular bases of such novel presentation. To this purpose, we first identified the specific mutation responsible for the occurrence of PHPT-1a. Then, we searched for possible pathogenic copy number variants (CNVs) or loss of heterozygosity (LOH) that could suggest the presence of homozygous damaging variants due to consanguinity/uniparental disomy. Finally, we performed a genetic analysis by clinical exome sequencing (CES) in search of alterations in genes known to be related to the pathogenesis of tooth agenesis.

## 4. Materials and Methods

After obtaining written informed consent, genomic DNA samples of the proband and her mother were extracted from peripheral blood lymphocytes, using the DNeasy Blood & TissueKit and QIAamp DNA Blood Mini Kit according to the manufacturers’ instructions. The coding regions and boundaries of flanking introns (±25 nucleotides) of the GNAS gene (Refseq: NM_000516.4) have been analyzed through Sanger sequencing (Applied Biosystems SeqStudio Genetic Analyzer, Thermofisher), using owner-designed primers (Table 1).

The identified variants have been evaluated, based on evidence from the scientific literature, and classified according to the criteria of the American College of Medical Genetics and Genomics (ACMG). Only those predicted to alter the protein and with a minor allele frequency, (MAF) < 0.01, were considered.

We also performed Affymetrix Single Nucleotide Polymorphism (SNP) array in our patient, to detect possible pathogenic copy number variants (CNVs) or loss of heterozygosity (LOH) that could suggest the presence of homozygous damaging variants due to consanguinity/uniparental disomy [9]. Genomic screening for CNVs was performed using a SNP array platform (Cytoscan HD, Thermo Fisher Scientific, Waltham, MA, USA), following the manufacturer’s recommendations, and analyzed with ChAS software (v4.1; Thermo Fisher). A total of 270 healthy controls belonging to the International HapMap Project were used as a reference sample in data analysis (Thermo Fisher). Called CNVs were represented by at least 25 contiguous probes and 75 kb as minimum size, and were classified according to the American College of Medical Genetics (ACMG) recommendations [10]. Moreover, all CNVs represented by at least 5 contiguous probes and laying within a candidate disease gene (OMIM, Online Mendelian Inheritance in Man; https://www.omim.org/ accessed on 4 November 2022) were considered.

Runs of Homozygosity (ROH) analysis of autosome chromosomes was performed using the SNPs and filtered considering 3 Mb of length as a minimum size [11]. CES was carried out using the TruSight One Sequencing Panel (Illumina, San Diego, CA, USA) according to the manufacturer’s instructions. The panel covers 4813 disease-associated genes. Targeted exonic regions underwent paired-end sequencing on an Illumina platform using a NextSeq 500 sequencing system (NextSeq High Output Kit, 300 cycles). The data analysis variant was carried out with the Illumina Variant Studio software v3.0 and BaseSpaceVariant Interpreter Beta (Illumina). Detected variants were annotated and filtered based on information of functional prediction (e.g., Polyphen2, SIFT, REVEL), disease association (e.g., ClinVar, HGMD, OMIM and GWAS) and population allele frequency (e.g., dbSNPs, ALFRED). Variant filtering was restricted to high quality variants in known pathogenic genes related to tooth agenesis (HP:0009804).

The main genes investigated were *AXIN2, EDA, LRP6, MSX1, PAX9, WNT10A, WNT10B, BMP4, DKK1, EDAR, EDARADD, GREM2, KREMEN1, LTBP3* and *SMOC2* [12,13]. Phenomizer algorithm was also utilized to semantically match the patient’s clinical features (HPO terms) to known disease–gene associations. Moreover, the Exomiser software was used to prioritize candidate variants based on gene–disease association, pathogenicity variant, and genetic algorithm based on a freely database. However, the CES analysis did not identify any pathogenic variants in investigated genes; instead, it confirmed the presence of the c.623_624dup pathogenic variant in the *GNAS* gene, which was prioritized with a significant *p*-value (*p* > 0.001) by the Exomiser tool when clinical phenotypes of PHPT-1a were set up in bioinformatic analysis tool.

## 5. Results

Genetic testing revealed the pathogenic insertion c.624dup, p.(Glu209*), in the exon 8 of the *GNAS* gene, leading to premature termination codon and truncated protein (Figure 4).

Segregation analysis confirmed the maternal origin of the identified variant. The SNP array did not identify any clinically significant deletions or duplications. However, it revealed ~1.75% autosomal homozygosity across multiple chromosomes, affecting a total of ~664 Mb (blocks ≥3 Mb), higher than the general population (Figure 5).

This level of homozygosity is consistent with a close parental relationship or more distant relatedness in an isolated population. In order to investigate the presence of possible damaging variants in tooth agenesis genes, including *AXIN2, EDA, LRP6, MSX1, PAX9, WNT10A, WNT10B, BMP4, DKK1, EDAR, EDARADD, GREM2, KREMEN1, LTBP3* and *SMOC2* genes, or in recessive genes located in the LOH regions and related to the proband’s phenotype, a CES was performed in the patient. No damaging mutations emerged in all those genes analyzed by CES, with the exception of the already-known c.624dup pathogenic variant in *GNAS*.

## 6. Discussion

We describe here, for the first time, a new clinical feature of PHPT-1a in a patient carrying the damaging variant c.624dup in the *GNAS* gene. Although this variant has never been described to date, neither in patients with *GNAS*-related diseases nor in the general population, null variants in *GNAS* are predicted to be pathogenic, as loss of function of this gene is a known mechanism of disease. The same variant was also detected in the asymptomatic mother. The absence of clinical manifestations in the older woman may be explained by the complex imprinted expression pattern of *GNAS*. Indeed, this gene produces paternally (XLAS), maternally (NESP55) and biallelically (Gsα) expressed transcripts from alternative promoters in the 5′-UTR. However, in the proximal renal tubule, adenohypophysis and thyroid, the expression of Gsα is exclusively maternal. Thus, damaging variants in *GNAS* on the maternal allele cause multihormonal resistance syndrome with AHO, while damaging variants in the *GNAS* paternal allele are associated only with isolated AHO (also known as pseudo-pseudo hypoparathyroidism). Realistically, the mother, not affected by PHPT-1a, has the pathogenic variant on the paternal allele, while, in the proband, the same variant, inherited from the mother, is the cause of her complex phenotype. The oral manifestations found in patients with parathyroid deficiency had been initially recognized by Gottlieb in 1920 [14]. In 1956, Hinrichs confirmed that hypoparathyroidism was associated with a delayed eruption and affected both matrix formation and calcification [15]. Pseudohypoparathyroidism is a disease very similar to idiopathic hypoparathyroidism, with almost identical clinical, radiographic, and histological dental manifestations. Therefore, the manifestations found in patients with PHPT-1a are primarily a late tooth eruption and/or aplasia or hypoplasia of the dental enamel [16,17]. However, it has been reported that dental manifestations of AHO are seldom sought after and they have only occasionally been described in a few case reports [18,19,20,21,22]. Recently, four patients presenting PHPT-1a and dental alterations have been described [23]. All patients exhibited dental anomalies, class III malocclusion with maxillary retrusion, and a copper beaten appearance of the skull. Treatment of hypocalcemia with supplementation of vitamin D and calcium improved the medical condition. However, the occurrence of complete pseudo-anodontia in a patient with PHPT-1a has not been reported so far and it is not even reported among the non-classic features of this disease [24]. The pathogenesis of tooth un-eruption in our patient is not clear. Several hypotheses may be postulated. The complete absence of tooth eruption could be due to prolonged hypocalcemia, or to the high level of serum PTH or of serum parathyroid hormone-related protein (PTHrP). In this regard, the importance of early diagnosis of PHPT-1a, when calcium serum levels are still in the normal range, has been recently emphasized to avoid the severe hypocalcemic symptoms, such as seizures, paresthesia, and tetany [25]. The relevance of PTHrP, as well as of the PTH/PTHrP receptor (PPR), in tooth eruption has been known for some time [26]. In particular, loss-of-function mutations in the PPR gene are responsible for the cessation of tooth eruption, causing primary failure of tooth eruption [27,28,29,30]. This was also confirmed by using a mouse model harboring a specific deletion of the PTHrP [31]. In this study, the loss of PTH-1r within PTHrP-expressing cells produced a dramatic periodontal and root phenotype. A deletion of signaling starting 3 days after birth, which is the start of the intraosseous phase of eruption, resulted in a severely underdeveloped periodontal ligament (PDL), as shown by the loss of the periostin marker. The acellular cementum, the layer of mineralized tissue that covers most of the root, was replaced with cellular cementum. The ultimate phenotype was a failure of the molars to emerge into the oral cavity (68% of first molars in knockout mice failed to erupt). However, no relationship between plasma PTHrP levels and failure of tooth eruption, as well as other dental manifestations of PHP, was found by others [32]. The mechanism leading to an uncompleted eruption process in our patient might be due to: (i) the lack of root formation and development that has long been considered the force responsible for eruption; (ii) the lack of biophysical traction forces exerted by the PDL; (iii) the incorrect tooth position that could hinder the eruption path of some teeth; and, finally (iiii) the extreme bone density of some areas that could represent an obstacle to dental progression. Molecular tests ruled out damaging chromosomal and genetic variants involving teeth agenesis-related genes. We cannot exclude the presence of other genetic defects not investigable with the available assays (e.g., mutations located in regulatory or non-coding regions), or alterations in the expression of genes as the results of the thyroid hormone resistance. In this regard, it has been reported that thyroid hormones promote osteoblast differentiation via the BMP/Smad signaling pathway [33] and, in particular, *BMP2* gene expression in the dental follicle is considered essential for tooth eruption [34,35]. The absence of mutations in the teeth agenesis-related genes indicate that the *GNAS* null mutation identified in our patient could play a pivotal role in determining her peculiar phenotype. We believe that this mutation, associated with the delay in the treatment of hypocalcemia, could be responsible for the occurrence of complete pseudo-anodontia, in addition to the other manifestations of the disease.

## 7. Conclusions (Learning Points)

1.Dental manifestations of AHO are seldom sought after and they have only occasionally been described in a few case reports.2.Complete pseudo-anodontia, defined as the clinical, but not radiographic, absence of all teeth, due to a failure in their eruption, has, so far, never been reported in patients with PHPT-1a.3.In our patient, complete pseudo-anodontia may be the result of long-standing, untreated hypocalcemia.4.Early diagnosis and treatment of hypocalcemia in patients affected by PHPT-1a is of paramount importance in order to avoid severe hypocalcemic symptoms, such as seizures, paresthesia, and tetany, and to allow physiological tooth formation and eruption.

## Figures and Tables

**Figure 1 diagnostics-12-02997-f001:**
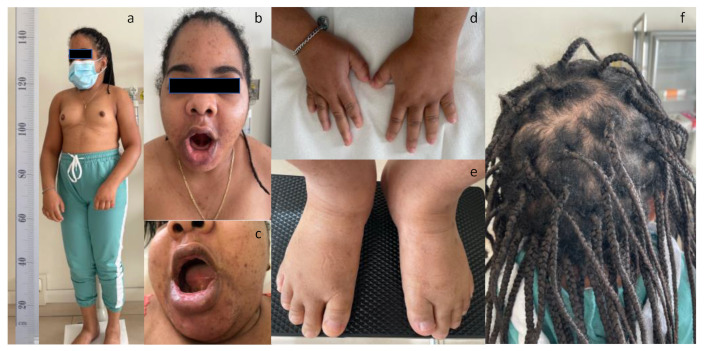
(**a**) Clinical characteristics of the patient with short stature. Closeup of the face (**b**) of the mouth (**c**), of the hands (**d**), of the feet (**e**) and of the hair (**f**).

**Figure 2 diagnostics-12-02997-f002:**
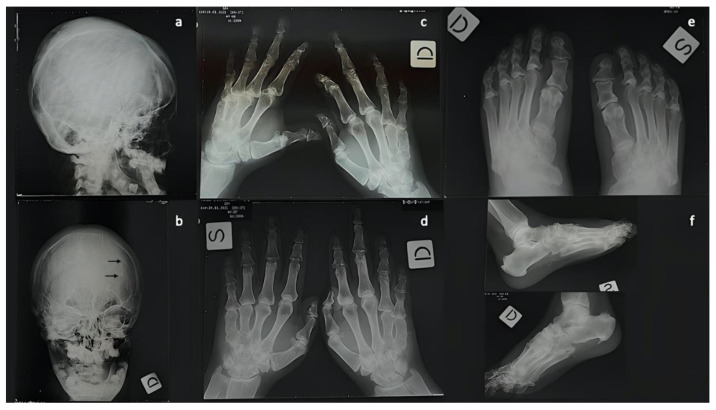
Radiologic findings of the patient. (**a**,**b**) X-ray of the skull, (**c**,**d**) X-ray of the hands, (**e**,**f**) X-ray of the feet.

**Figure 3 diagnostics-12-02997-f003:**
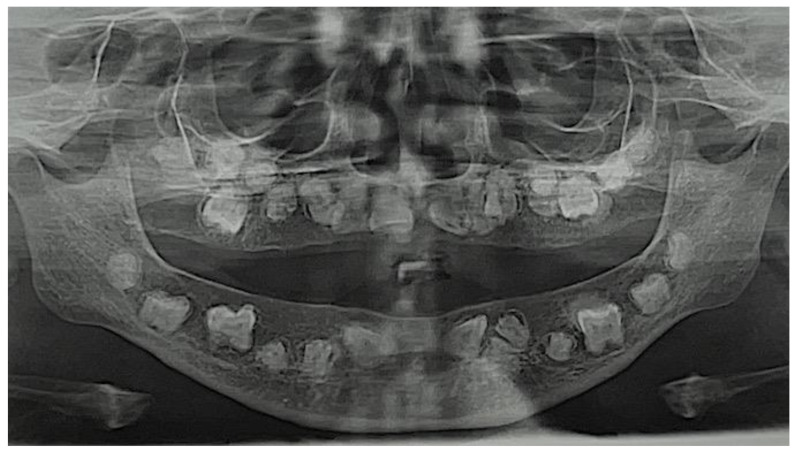
Dental panoramic radiography.

**Figure 4 diagnostics-12-02997-f004:**
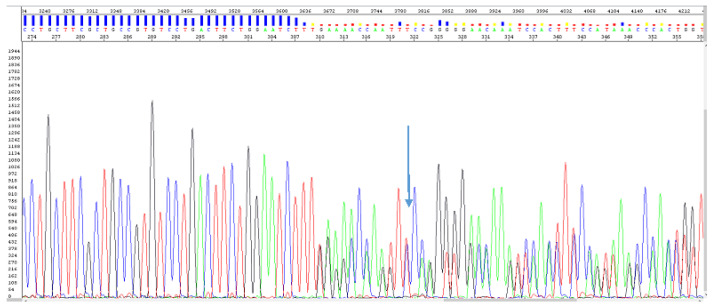
Sanger sequencing of the *GNAS* exon 8, showing (blue arrow) the heterozygous variant NM_000516.4: c.624dup (p.Glu209*).

**Figure 5 diagnostics-12-02997-f005:**
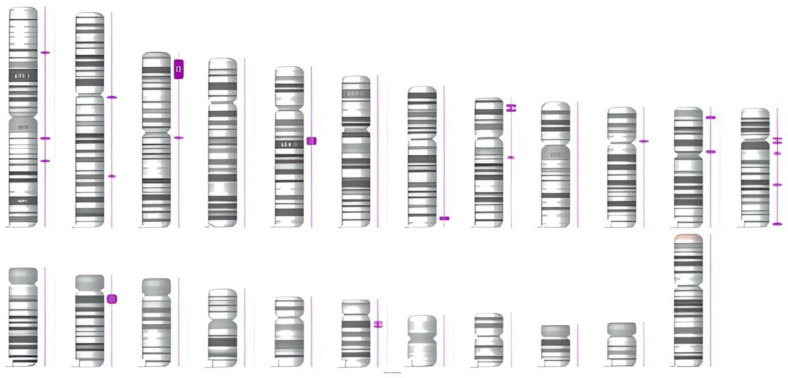
Loss of heterozygosity regions identified with SNP array.

**Table 1 diagnostics-12-02997-t001:** Primers used for the amplification and analysis of the *GNAS* gene.

Gene	Exon	Primers	Sequence
*GNAS*	EX1	1F	5′ TCCTTGCCGAGGAGCCGAG 3′
*GNAS*	EX1	1R	5′ CACAGACAGAGCCCGCGAAC 3′
*GNAS*	EX2	2F	5′ GTCAAGGAAAGTTGCAAGTCTG 3′
*GNAS*	EX2	2R	5′ AGAGCCCTTCCCAGGATTTTC 3′
*GNAS*	EX3	3F	5′ TGGCTGATGGTTGAGGAATGTA 3′
*GNAS*	EX3	3R	5′ TATGCCAATATGGCTGATGGTC 3′
*GNAS*	EX4+5	4+5F	5′ GAACCCACAACTCCCTGAAGA3′
*GNAS*	EX4+5	4+5R	5′ TTCCTATATGGACACTGTGCTC 3′
*GNAS*	EX6	6F	5′ GTGTCGGTCACATAGGGAACT 3′
*GNAS*	EX6	6R	5′ CAGTGGGGTAACTGGTTGGC 3′
*GNAS*	EX7+8	7+8F	5′ GGGACGGTCACTTCCGTTGA3′
*GNAS*	EX7+8	7+8R	5′ ACAGCTGGTTATTCCAGAGGG 3′
*GNAS*	EX9+10	9+10F	5′ CCCTCTGGAATAACCAGCTGT 3′
*GNAS*	EX9+10	9+10R	5′ CTTGGGAGAAGCGCGCTTTC 3′
*GNAS*	EX11	11F	5′ AGGAGGCCCTGGTCTGCAC 3′
*GNAS*	EX11	11R	5′ ATGGTTTGGTGGTGGGAGGG 3′
*GNAS*	EX12+13	12+13F	5′ AGGGTTTTGAAGACTTCAGGAG 3′
*GNAS*	EX12+13	12+13R	5′ GCCCTATGGTGGGTGATTAACT 3′

## Data Availability

Not applicable.

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
