# Peer review of "Complete Pseudo-Anodontia in an Adult Woman with Pseudo-Hypoparathyroidism Type 1a: A New Additional Nonclassical Feature?"

_diagnostics, 2022, doi:10.3390/diagnostics12122997_

Round 1
Reviewer 1 Report
The paper "Complete Pseudo-anodontia in an adult woman with Pseudo-hypoparathyroidism type 1a: a new additional nonclassical features?" is of some interest.
The abstract is clear and concise.
The introduction is clear and provides sufficient background.
The case presentation is well described with some figures that help the reader in the description.
Materials and methods are essential and summarized but quite clear
Results are clearly presented with the help of two figures.
The conclusions of the case report are clear.
The cited references are relevant for the paper
Author Response
Thank you for your appreciation to our manuscript.
Reviewer 2 Report
Dear editor in chief,
In a study entitled "Complete Pseudo-anodontia in an adult woman with Pseudo-hypoparathyroidism, type 1a: a new additional nonclassical features?" the authors tried to report the novel association of Pseudo-hypoparathyroidism type 1a with alopalopecia. This article is supposed to be a case report study; however, the style of writing and making conclusion is a little bit different from the primary purpose of the study; follows are my recommendations to improve the article:
1. The article needs to rewrite with an appropriate structure with explicit hypotheses and aims.
2. English proficiency should be improved to make the article more understandable.
3. In Introduction section, authors need to add further information about background and hypothesis.
4. For oral manifestation of the patient I recommended a picture which can shows the paragraph that authors mentioned (Page 3, paragraph 2)
Author Response
We are really grateful to Reviewer 2 for the valuable suggestions. We are still convinced that the article is indeed a case report because it started from the observation of the peculiar phenotype observed in our patient affected by a rare disease. In the manuscript preparation we followed the guidelines reported in the literature [Guidelines To Writing A Clinical Case Report. Heart Views. 2017 Jul-Sep; 18(3): 104–105. doi: 10.4103/1995-705X.217857: 10.4103/1995-705X.217857]. The structure of the manuscript, in fact was based on the following scheme and components (abstract, introduction, case presentation, and discussion) and it is a detailed report of the symptoms, signs, diagnosis, treatment, and follow-up of a single patient. In addition, it is focused on the unusual observation of the association of the typical clinical picture of Pseudo-hypoparathyroidism type 1a (PHPT-1a) with complete pseudo-anodontia and alopecia. However, we agree with the Reviewer 2 that the manuscript contains complex experiments and a deep insight into the molecular genetic background of our patient that enriches the description of the case and gives additional relevant results. For this reason, the manuscript includes also additional sections (Materials and Methods and Result) to explain the way we performed our search with the aim to look for possible gene alterations associated with the known genetic GNAS null mutation. Unless the reviewer is asking to modify the structure of manuscript and to change it in an original research study, we would like to continue to consider it as a case report.
However, we accept the suggestion of the reviewer 2 and we added a sentence in the abstract, in the introduction, as well as and in the New Section named Aim of the study, to describe the hypothesis and the aim and to make the article more understandable.
Unfortunately, we are not able to add an additional figure for the visualization of the oral manifestation of the patient, as asked by the reviewer 2, because she returned to her Country of origin and she has no plan to come back soon to Italy. We specified this in the treatment outcome and follow-up paragraph.
We have re-examined the manuscript to provide a better and higher quality English content.
All the changes in the text are highlighted in yellow.